# Development and Applicability Evaluation of Damage Scale Analysis Techniques for Agricultural Drought

**Youngseok Song [1], Jingul Joo [2], Hayong Kim [3] and Moojong Park [4,***

[1] Department of Fire and Disaster Prevention, Konkuk University, Chungju 27478, Republic of Korea; ys-song@kku.ac.kr
[2] Department of Civil Environmental Engineering, Dongshin University, Naju 582452, Republic of Korea; jgjoo@dsu.ac.kr
[3] Department of Construction Environment Research, Land and Housing Institute, Dajeon 34047, Republic of Korea; civilkhy@lh.or.kr
[4] Department of Aeronautics and Civil Engineering, Hanseo University, Seosan 31962, Republic of Korea
* Correspondence: mjpark@hanseo.ac.kr; Tel.: +82-416-601-051

**Abstract:** In recent years, the intensity and frequency of droughts have been increasing with the advent of the climate crisis. Agricultural droughts have a significant economic and social impact. Agricultural drought is not only a natural disaster but also leads to food security threats and reduced economic activities, such as decreased productivity. Therefore, it is very important to specify the scale of agricultural drought and quantitatively estimate the economic damage. In this study, we developed an analytical methodology to quantitatively assess the economic damage of agricultural drought and estimated the damage of agricultural drought in 2018 and 2019 for the Republic of Korea. The 2018 agricultural drought was estimated to have caused USD 4.438 million in damage cost and USD 5.180 million in recovery cost. The 2019 drought was less damaging than the previous year, with an estimated damage cost of USD 286,000 and recovery costs of USD 218,000. The results suggest that the economic impact of agricultural drought varies by region depending on the frequency and intensity of the drought and confirm the importance of regional strategies for effective drought management and response. The impacts of agricultural drought go beyond short-term agricultural losses and lead to long-term economic burdens. Therefore, the results of this study are expected to be used as a basis for understanding the impacts of agricultural drought on national economies and for developing policies and strategies to minimize impacts.

**Keywords:** agricultural drought; damage cost; damage factors; damage scale; recovery cost; drought





## 1. Introduction

The occurrence of drought causes complex damage, such as food security threats, reduced productivity, and economic losses due to lack of rainfall. In particular, the frequency and intensity of droughts are expected to increase as the phenomenon of climate change intensifies. Agricultural drought is the type of drought that is most sensitive to the effects of climate. Increased temperatures and decreased rainfall due to climate change have a significant impact on crop growth by causing agricultural droughts, ultimately leading to reduced yields [1]. Agricultural droughts have a significant impact on crop growth and ultimately result in reduced yields. This has a negative economic impact as it leads to lower farm incomes and higher food prices. The effects of agricultural droughts are not only manifested in direct agricultural losses but also in reduced economic activity across the board [2,3].

Estimating the quantitative amount of damage caused by agricultural drought is a very complex research field. The impact of drought increases or decreases depending on various factors, such as regional characteristics, the presence of mitigation facilities, and proactive policy support [4]. In addition, quantitative estimates of damage cost and

recovery cost are necessary to develop effective drought prevention and response strategies when damage occurs.

Previous research on agricultural drought has been conducted using a variety of research methods targeting regions around the world. The research method presented application methods to respond to drought through climate impact, economic assessment, risk assessment, prediction models, drought index, policy analysis, water resource management and water economics. The impact of climate change on agricultural productivity in the United States and China was analyzed, and crop yield was calculated through hydro-economic modeling [5–11]. Economic assessment and policy analysis of agricultural drought has been studied in a variety of countries, including Spain, Australia, China, and Iran. They have assessed not only the direct economic losses caused by agricultural drought, but also the potential economic impacts of drought management policies and strategies [12–16]. From a water economics perspective, it has been shown that drought events have a negative impact on GDP; industrial output; employment; and key macroeconomic variables, such as trade balance and household consumption. The importance of water economics is emphasized by the fact that the absence of drought mitigation mechanisms can lead to significant socioeconomic losses [16,17].

Studies on agricultural drought risk assessment and water resource management strategies have been conducted in various countries. Agricultural drought risk analysis has applied a variety of analytical methods, including satellite data, hydrological and economic modeling, and policy assessment and strategy development to assess impacts and propose responses [17–38]. Hydro-economic modeling has been used to assess the impacts of agricultural droughts in Spain, the United States, and China. Water management and response policies have also been proposed to minimize economic losses [29,30,32–35]. In Nigeria, Spain, and Kenya, the economic value of drought forecasting information was assessed, and policy and technical measures were proposed to enhance farmers' adaptive capacity [37,38].

The development of drought prediction models utilized precipitation data and climate models for the UK and South Africa. These models predicted both the frequency and intensity of droughts, which were then evaluated for their impact on agricultural productivity [39–41]. Drought indicators were also developed to identify different cycles of drought and predict the likelihood of future drought events [11].

Policy analysis on agricultural drought has been conducted in Spain and Nigeria to evaluate the link between drought indexes and insurance schemes. The feasibility and economic valuation of climate change insurance schemes have been analyzed to suggest policy and technical measures to strengthen farmers' adaptive capacity [33–35,38,42]. The impacts of agricultural drought on water resource management have been analyzed. Hydro-economic modeling has been used to analyze water scarcity and agricultural pollution and to propose countermeasures to minimize economic losses through interregional water transfers [29,30,32,36].

The economic impacts of previous studies on agricultural droughts were qualitatively evaluated proportionally to agricultural droughts, regional drought severity, and economic impacts. Therefore, this study aims to develop an analytical method of estimating the damage cost and recovery cost for agricultural drought in Korea. The methodology consists of developing an equation for quantitative estimation of damage cost and recovery cost. The analytical methodology selects damage factors based on previous studies on agricultural drought and damage data. The applicability of the developed methodology was evaluated based on the damage history of the agricultural drought that occurred in 2018 and 2019.

## 2. Materials and Methods

### 2.1. Selection of Damage Factors for Agricultural Drought

The damaging effects of agricultural drought are not limited to direct impacts on crop production, but also include socioeconomic impacts. In order to quantitatively assess the damage of agricultural drought, we investigated previous studies that can estimate the damage

of drought in Korea and abroad (Table 1) [43–45]. The US used the SWAP model to estimate the damage scale of agricultural drought, and Japan applied the drought damage cost calculation method. In Korea, a damage cost calculation method that considers agricultural productivity—such as delayed planting, paddy drying, and pest infestation—was presented.

**Table 1.** Studies of agricultural drought damage estimation.

| Country | Reports | Damage Calculation Factors |
|---|---|---|
| United States | SWAP version 4 (2017) | Economic factors, drought performance, costs and revenues, water restrictions and surcharges, acreage, and crop failures |
| Japan | Agricultural Crop Insurance Loss Assessment Outline (2022) | Agricultural water savings rate, agricultural water withdrawals, agricultural damage cost, agricultural damage days, agricultural damage rate, agricultural value added tax |
| Republic of Korea | National Drought Information Statistical Compendium (2020) | Damage area (rice paddy drying up and field withering), water support (dams, water supply, complementary water, national underground network, water trucks, bottled water), water source development (tube wells, downhole excavation, water membrane, field spring development, simple pumping station, water storage, dredging), equipment support (water pumping machine, drilling machine, water bag, excavator water supply vehicle, water transfer lake), and personnel support (residents, government officials, military police, institutional organizations, public works) |

In the United States and Japan, factors and methods for calculating direct drought damages have been proposed. However, Korea does not have a formalized methodology for analyzing drought. Only data on damage factors and recovery factors for drought damage are presented. In Korea, the impacts of drought are managed as a disaster response, with damage and recovery as a part of the disaster response process. Therefore, this study distinguishes between damage and recovery factors for agricultural drought.

Agricultural drought losses were categorized into crop product damage, livestock product damage, and fisheries product damage (Table 2). Crop product damage was measured by damage area, total crop products income, and damage ratio. livestock products damage was measured by the livestock produce and livestock total income. Fisheries product damage was measured by the fisheries' mortality and the fisheries' production costs.

**Table 2.** Selection of damage factors for agricultural drought.

| Type | Damage Factors |
|---|---|
| Crop product damage | − Damage area (ha), total crop product income (USD/10 ha), damage ratio (unharvestable area/ planted area of crop at time of damage x 100 (%)) |
| Livestock product damage | − Livestock produce (count or kg), livestock total income (USD/kg) |
| Fisheries product damage | − Fisheries' mortality (kg), fisheries production cost (USD/kg) |

The recovery factors for agricultural drought are agricultural drought support, water support, and equipment and labor support (Table 3). Agricultural drought support was selected as a recovery factor for Crop substitute crops, young livestock, and fisheries fry. Water support was selected as a recovery factor for water supply during agricultural droughts, including dam water, tap water, barrage water, groundwater, water wagon, and water bag. Equipment and manpower support were selected for water pump, water hose, public official, military and police, and general public.

**Table 3.** Selection of recovery factors for agricultural drought.

| Type | Recovery Factors |
|------|------------------|
| Agricultural drought support | – Crop substitute crops (USD/10), young livestock (USD/count, kg), fisheries fry (USD/kg) |
| Water support | – Dam water ($m^3$), tap water ($m^3$), barrage water ($m^3$), Groundwater ($m^3$), water wagon (count), water bag (count) |
| Equipment and labor support | – Water pump (count), water hose (count), public official (person), military and police (person), general public (person) |

### 2.2. Development of Damage Cost Analysis Technique for Agricultural Drought

The agricultural drought damage cost analysis methodology developed a calculation formula based on the damage factors investigated in previous studies. The agricultural drought damage cost analysis methodology calculates the sum of the damage cost of crop products, damage cost of livestock products, and damage cost of fisheries products, as shown in Equation (1).

$$d\_ad = cp + lp + fp \tag{1}$$

where $d_{ad}$: damage cost of agricultural drought, cp: damage cost of crop products, lp: damage cost of livestock products, fp: damage cost of fisheries products.

The amount of crop damage cost is equal to Equation (2) for damage area of crop products (apa), damage ratio (dr), and production cost of crop products (ccp). Here, the damage ratio (dr) of crop damage is calculated as the ratio of the damaged area that cannot be harvested to the crop cultivation area in the year of damage, as shown in Equation (3). The damage cost of livestock products is calculated as shown in Equation (4) for damage mortality of livestock products (mpm) and production cost of livestock products (clp). Damage cost to fisheries products is calculated as follows: damage mortality of fisheries products (mfp) and production cost of fisheries products (cfp).

$$cp = \sum_{cp=1}^{n} \left[ \left( apa_{cp} \times dr \right) \times ccp_{cp} \right] \tag{2}$$

$$dr = \frac{\text{damaged area that cannot be harvested}}{\text{crop cultivation area in the year of damage}} \times 100 \tag{3}$$

$$lp = \sum_{lp=1}^{n} \left( mpm_{lp} \times clp_{lp} \right) \tag{4}$$

$$fp = \sum_{fp=1}^{n} \left( mfp_{fp} \times cfp_{fp} \right) \tag{5}$$

where $cpa_{cp}$: crop products damage area (ha), $cpc_{cp}$: crop products production cost (USD/ha), dr: damage ratio (%), $mpm_{lp}$: damage mortality of livestock products (count, kg), $clp_{lp}$: production cost of livestock products (USD), $mfp_{fp}$: damage mortality of fisheries products (kg), $cfp_{fp}$: production cost of fisheries products (USD).

The total income for crops was calculated by province for 11 crops from the Crop Income Data Collection (Rural Development Administration). For livestock products, the total income of seven livestock products from Statistics on Livestock Production Costs (Statistics Korea) was used. Fishery products were calculated by applying the production value of 49 types of marine fisheries and 35 types of inland fisheries from the Fishery Production Trends Survey (Statistics Korea) and Statistics by Fishery Type (National Statistics Portal). The calculation criteria for agricultural drought damage factors are as shown in Table 4.

**Table 4.** Calculation standard for damage cost by agricultural drought damage factor.

| Type | Calculation Standard |
|---|---|
| Crop products damage cost | − Gross income by crop from the Agricultural Income Data Sheet (Rural Development Administration) |
| Livestock products damage cost | − Total income by livestock product from Statistics on Livestock Production Costs (Statistics Korea) |
| Fisheries products damage cost | − Amount of production by variety from Fisheries Production Trend Survey (Statistics Korea) |

*2.3. Development of Recovery Cost Analysis Technique for Agricultural Drought*

The agricultural drought recovery cost methodology developed an equation based on recovery factors researched in previous studies. The methodology calculates the recovery cost for agriculture drought support (ads), water support (ws), and equipment and manpower support (ems), as shown in Equation (6).

$$r_{ad} = \sum_{a=1}^{3} ads_a + \sum_{w=1}^{6} ws_w + \sum_{em=1}^{5} ems_{em} \tag{6}$$

where $r_{ad}$: recovery costs of agricultural drought, $ads_a$: agriculture drought support, $ws_w$: water support, $ems_{em}$: equipment and manpower support.

Agricultural drought support consisted of crop replanting, livestock piglets, and fishery fish stocking support. Water support consisted of dam water support, water supply network support, supplementary water support, national underground water network support, water trucks, and water bags. Equipment and personnel support consisted of water pumpers, water transfer lakes, government officials, military police, and the general public. The recovery factors of the agricultural drought recovery cost analysis method are shown in Table 5.

**Table 5.** Recovery factors for each parameter of agricultural drought recovery cost.

| Type | Parameter |
|---|---|
| Agricultural drought support ($ads_a$) | $ads_1$: crop substitute crops, $ads_2$: young livestock, $ads_3$: fisheries fry |
| Water support ($ws_w$) | $ws_1$: dam water, $ws_2$: tap water, $ws_3$: barrage water, $ws_4$: groundwater, $ws_5$: water wagon, $ws_6$: water bag |
| Equipment and labor support ($ems_e$) | $ems_1$: water pump, $ems_2$: water hose, $ems_3$: public official, $ems_4$: military and police, $ems_5$: general public |

Seed and seedling costs from the Agricultural Income Data Book (Korea Agriculture and Rural Development Administration) by crop were used for agricultural drought support. For livestock piglet support, the cost of piglets from the Statistics on Livestock Production Costs (Statistics Korea) was used. Fisheries and Aquaculture Fry support was based on support discrimination. For water support, dam water, municipal water, and complementary water are based on the unit price announced by the Water Resources

Corporation. For groundwater, we applied the unit price announced by local governments. The unit price of the Korea Water Truck Association was used for water trucks, and the unit price of water bags was used for local governments. The calculation criteria applied to the agricultural drought recovery factor is as shown in Table 6.

**Table 6.** Agricultural drought recovery factors by recovery cost.

| Type | | Calculation Standard |
|---|---|---|
| Agricultural drought support | Crop substitute crops | − Costs by crop from the Agricultural Income Data Sheet (Korea Agricultural Research Service) |
| | Young livestock | − Piglet cost by livestock product from Livestock Production Cost Statistics (Statistics Korea) |
| | Fisheries fry | − Fish prices in the municipality |
| Water support | Dam water | − Rates published by the water utility |
| | Tap water | − Rates published by the water utility |
| | Barrage water | − Rates published by the water utility |
| | Groundwater | − Fares announced by the subway system |
| | Water wagon | − Rates published by the Korea Water Distribution Association |
| | Water bag | − Fares announced by the subway system |
| Equipment and labor support | Water pump | − Based on Construction Machinery Expense Calculation Table (Korea Association of Construction Engineers) |
| | Water hose | − Fares announced by the subway system |
| | Public official | − Based on the Government Overtime Pay Unit Rate |
| | Military and police | − Based on Overtime pay unit rate for military sergeants and police sergeants |
| | General public | − Public Works Wage Standards |

## 3. Results

### 3.1. Assessment of Applicability of Damage Cost from Agricultural Drought

3.1.1. Calculating Agricultural Drought Damages Cost for 2018

In this study, we estimate the agricultural drought damage cost for 17 provinces nationwide in 2018. The data of agricultural drought damage from the National Drought Information Statistical Compendium in 2018 (2020, Ministry of Public Administration and Security) were utilized [45]. The methodology for analyzing the damage cost to crops, livestock, and fisheries products was developed in Section 2.2. However, the damage to crops in 2018 was only counted for agricultural drought. In some articles and reports, the status of damage to livestock products and fisheries products was presented. However, this study used official data published by the government and damage status data by administrative district. The types of damage caused by agricultural drought are presented in terms of area affected by rice paddies drying up and fields wilting. The value of agricultural drought damage cost was calculated for crops grown in paddy fields and fields. However, in order to apply Equation (3), both damaged areas that cannot be harvested and crop cultivation areas in the year of crop damage in 2018 must be presented. However, the damage data currently provided by the government only presents the damaged area, so

the areas of damaged area that cannot be harvested and crop cultivation area in the year of damage were applied equally.

The gross income by crop for paddy and field was calculated based on the gross income from the Agricultural Income Data Book in 2018 (2019, Korea Rural Development Administration) [46]. For paddies, the total income by province for rice was calculated by applying the second quartile of the total income by 119 crops in the field. For field crops, we used the second quartile of total income, as the variation in total income by province is large, which tends to cause overestimation of the average. The total income for paddy and field crops for the 17 provinces in 2018 is shown in Table 7.

**Table 7.** Production cost from crops products by province in 2018.

| Si/Do | Production Cost (USD/10 ha) | |
|---|---|---|
| | **Paddy** | **Field** |
| Seoul-si | 900 | 7550 |
| Busan-si | 900 | 2910 |
| Daegu-si | 900 | 3610 |
| Incheon-si | 900 | 3190 |
| Gwangju-si | 900 | 7100 |
| Daejeon-si | 900 | 8000 |
| Ulsan-si | 900 | 4120 |
| Sejong-si | 900 | 4100 |
| Gyeonggi-do | 860 | 2250 |
| Gangwon-do | 890 | 2270 |
| Chungcheongbuk-do | 870 | 1830 |
| Chungcheongnam-do | 980 | 2420 |
| Jeollabuk-do | 970 | 1750 |
| Jellanam-do | 850 | 1910 |
| Gyeongsangbuk-do | 910 | 1820 |
| Gyeongsangnam-do | 880 | 1980 |
| Jeju-do | 900 | 2070 |
| Average | 900 | 3460 |

The gross income from paddy by province was the same in the National Capital Region due to the small area of paddy cultivated, while the other nine provinces ranged from USD 850 to USD 980 per 10 ha. Gross income from fields ranged from USD 1750 to USD 7550 with an average of USD 3460 due to the different crops grown in each province.

In 2018, 10 out of 17 provinces in the country were affected by agricultural drought. Seoul, Daejeon, Sejong, Daegu, Ulsan, Busan, and Gyeongsangnam-do were excluded. The total amount of agricultural drought damage cost in 2018 was USD 4.438 million, with the largest amount of damage in Jeju Island (USD 1.793 million) and the smallest amount in Gwangju City (USD 7000). Figure 1 shows the status of agricultural drought damage by province.

Jeju-do, Gangwon-do, and Jeollanam-do are the regions with more than USD 400,000 in agricultural drought damage costs in 2018, while Gyeongsangbuk-do and Chungcheongnam-do are the regions with more than USD 200,000. Incheon, Chungcheongbuk-do, and Jeollabuk-do were the regions with more than USD 100,000, and Gwangju was the region with less than USD 100,000. In terms of crop damage cost, USD 230,000 was estimated for rice paddies, and USD 4.208 million was estimated for fields. The large difference between the damage cost to rice paddies and fields is due to the large difference in total income by province. The amount of agricultural drought damage cost by province in 2018 is shown in Table 8.

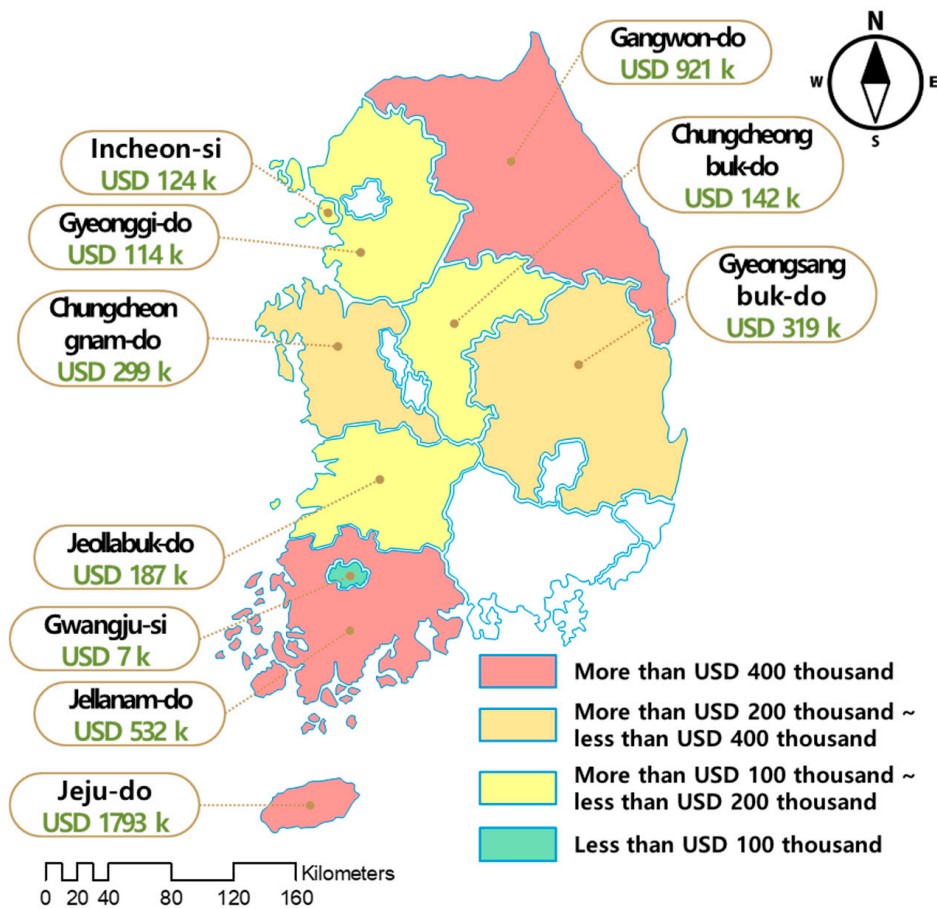

**Figure 1.** Damage cost from agricultural drought by province in 2018.

**Table 8.** Damage cost from agricultural drought by province and factor in 2018.

| Si/Do | Damage Cost (Thousands of USD) | | |
|---|---|---|---|
| | **Paddy** | **Field** | **Sum** |
| Seoul-si | 0 | 0 | 0 |
| Busan-si | 0 | 0 | 0 |
| Daegu-si | 0 | 0 | 0 |
| Incheon-si | 59 | 65 | 124 |
| Gwangju-si | 1 | 6 | 7 |
| Daejeon-si | 0 | 0 | 0 |
| Ulsan-si | 0 | 0 | 0 |
| Sejong-si | 0 | 0 | 0 |
| Gyeonggi-do | 17 | 97 | 114 |
| Gangwon-do | 52 | 869 | 921 |
| Chungcheongbuk-do | 3 | 139 | 142 |
| Chungcheongnam-do | 61 | 238 | 299 |
| Jeollabuk-do | 12 | 175 | 187 |
| Jellanam-do | 15 | 517 | 532 |
| Gyeongsangbuk-do | 10 | 309 | 319 |
| Gyeongsangnam-do | 0 | 0 | 0 |
| Jeju-do | 0 | 1793 | 1793 |
| Sum | 230 | 4208 | 4438 |

### 3.1.2. Calculating Agricultural Drought Damages Cost for 2019

For the agricultural drought that occurred in 2019, we calculated the damage cost for 17 provinces nationwide. The data of agricultural drought damage was utilized from the 2019 yr National Drought Information and Statistics (2021, joint ministries) [47]. As

in 2018, only crop damage data were collected for agricultural drought in 2019. In 2019, drought severity was not expected to be high, and only crop damage occurred among the agricultural drought damage factor. The types of damage caused by agricultural drought are presented in terms of the area of damage to rice fields and field withering. However, in order to apply Equation (3), both damaged areas that cannot be harvested and crop cultivation areas in the year of crop damage in 2019 must be presented. However, the damage data currently provided by the government only presents the damaged area, so the areas of damaged area that cannot be harvested and crop cultivation area in the year of damage were applied equally.

The gross income by crop for paddy and field was calculated based on the gross income from the 2019 Agricultural Income Data Book (2020, Korea Rural Development Administration) [48]. The gross income thresholds for paddies and fields were the same as in 2018. The gross incomes of paddies and fields for the 17 provinces in 2019 are shown in Table 9. The total incomes from paddy by province is the same for the National Capital Region and ranges from USD 790 to USD 950 per 10 ha for the other nine provinces. Gross income from fields was calculated by applying the second quartile of gross income from crops grown in each province and ranged from USD 2620 to USD 10,890 per 10 ha.

**Table 9.** Production cost from crops products by province in 2019.

| Si/Do | Production Cost (USD/10 ha) | |
|---|---|---|
| | Paddy | Paddy |
| Seoul-si | 900 | 6030 |
| Busan-si | 900 | 9250 |
| Daegu-si | 900 | 7000 |
| Incheon-si | 900 | 4890 |
| Gwangju-si | 900 | 8500 |
| Daejeon-si | 900 | 10,890 |
| Ulsan-si | 900 | 6370 |
| Sejong-si | 900 | 10,170 |
| Gyeonggi-do | 870 | 5180 |
| Gangwon-do | 910 | 3310 |
| Chungcheongbuk-do | 870 | 3280 |
| Chungcheongnam-do | 940 | 5970 |
| Jeollabuk-do | 930 | 3870 |
| Jellanam-do | 790 | 3670 |
| Gyeongsangbuk-do | 910 | 3360 |
| Gyeongsangnam-do | 950 | 3350 |
| Jeju-do | 900 | 2620 |
| Average | 900 | 5750 |

The 2019 agricultural drought occurred in 3 regions—Incheon, Chungcheongnam-do, and Gangwon-do—out of 17 provinces nationwide. The total amount of agricultural drought damage cost in 2019 was USD 286,000, with the largest amount of damage in Gangwon Province, which was estimated at USD 189,000. Figure 2 shows the status of agricultural drought damage in each province.

The year 2019 was a low-impact year for agricultural drought, with less than USD 200,000 in damages in 3 of the 17 provinces. Damage to paddy fields occurred in three provinces and damage to fields occurred in only Chungcheongnam-do. The value of agricultural drought damage cost was estimated at USD 35,000 in Incheon, USD 189,000 in Gangwon, and USD 62,000 in Chungcheongnam. The amount of agricultural drought damage cost by province in 2019 is shown in Table 10.

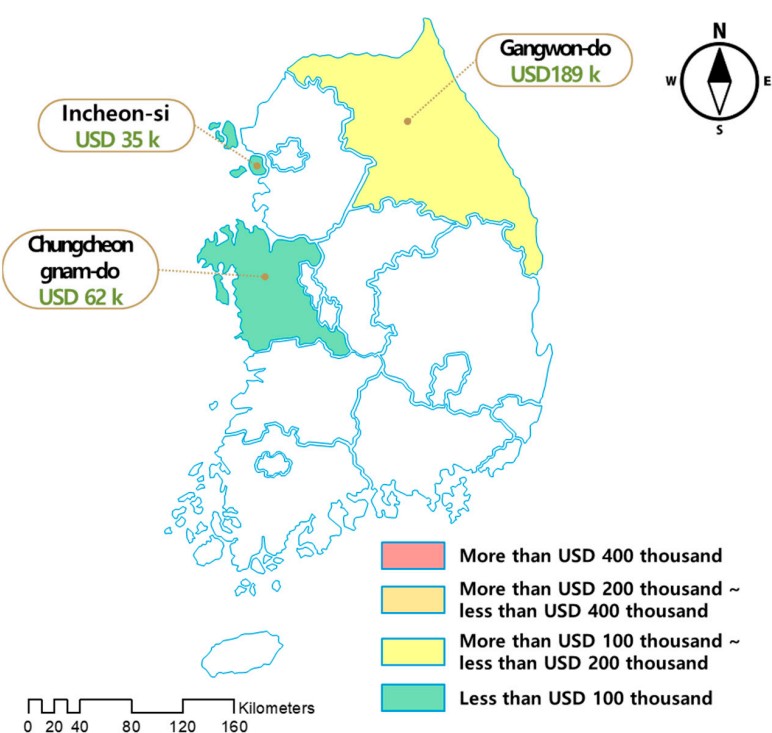

**Figure 2.** Damage cost from agricultural drought by province in 2019.

**Table 10.** Damage cost from agricultural drought by province and factor in 2019.

| Si/Do | Damage Cost (Thousands of USD) | | |
|---|---|---|---|
| | Paddy | Field | Sum |
| Seoul-si | 0 | 0 | 0 |
| Busan-si | 0 | 0 | 0 |
| Daegu-si | 0 | 0 | 0 |
| Incheon-si | 35 | 0 | 35 |
| Gwangju-si | 0 | 0 | 0 |
| Daejeon-si | 0 | 0 | 0 |
| Ulsan-si | 0 | 0 | 0 |
| Sejong-si | 0 | 0 | 0 |
| Gyeonggi-do | 0 | 0 | 0 |
| Gangwon-do | 189 | 0 | 189 |
| Chungcheongbuk-do | 0 | 0 | 0 |
| Chungcheongnam-do | 38 | 24 | 62 |
| Jeollabuk-do | 0 | 0 | 0 |
| Jellanam-do | 0 | 0 | 0 |
| Gyeongsangbuk-do | 0 | 0 | 0 |
| Gyeongsangnam-do | 0 | 0 | 0 |
| Jeju-do | 0 | 0 | 0 |
| Sum | 262 | 24 | 285 |

### 3.2. Assessment of Applicability of Recovery Cost from Agricultural Drought

3.2.1. Calculating Agricultural Drought Recovery Cost for 2018

In this study, we estimate the amount of agricultural drought recovery cost for 17 provinces nationwide in 2018. The recovery data of agricultural drought was utilized from the National Drought Information and Statistics 2018 (2020, joint ministries) [46]. Agricultural drought recovery cost can be calculated for a total of 14 recovery factors, including agricultural drought support, water support, equipment, and labor. In 2018, only crop damage was counted in the drought damages, so agricultural drought support was calculated for crop destruction.

The basis for calculating recovery cost for agricultural drought assistance, water assistance, and equipment and labor for the 2018 agricultural drought by state is shown in Table 11. Agricultural drought assistance was applied to crop failures based on the damage cost calculation. Crop replanting assistance is based on the cost of rice and field seedlings, averaged across provinces. In water support, dams, waterworks, and groundwater were applied based on the amount of water from the Water Resources Corporation. Water delivery vehicles and water bags were based on the amount from the National Water Truck Association. In Equipment and Labor, the average unit cost of equipment, water pumping stations and water transmission lakes, and labor, civil servants, military police, and public employees, was based on the civil service regulations.

**Table 11.** Calculation standard for agricultural drought recovery cost by province in 2018.

| Agriculture Drought Support | | Water Support | | Equipment and Manpower Support | |
|---|---|---|---|---|---|
| Type | Cost (USD/10 ha) | Type | Cost (USD/100 m³ or Count) | Type | Cost (USD/Count or Person) |
| Crop substitute crops | | Dam water | 4 | Water pump | 231 |
| Paddy | 15 | Tap water | 33 | Water hose | 23 |
| Fields | 315 | Barrage water | 33 | Public official | 64 |
| | | Groundwater | 55 | Military and police | 57 |
| | | Water wagon | 461 | General public | 46 |
| | | Water bag | 55 | | |

Agricultural drought recovery in 2018 occurred in 13 of the 17 provinces, with the exception of Seoul, Daejeon, Sejong, and Daegu. The total amount of agricultural drought recovery cost in 2018 was USD 5180 thousand, with Gyeonggi Province receiving the largest amount of USD 1242 thousand and Ulsan Metropolitan City receiving the smallest amount of USD 2 thousand. The agricultural drought recovery status of each province is shown in Figure 3.

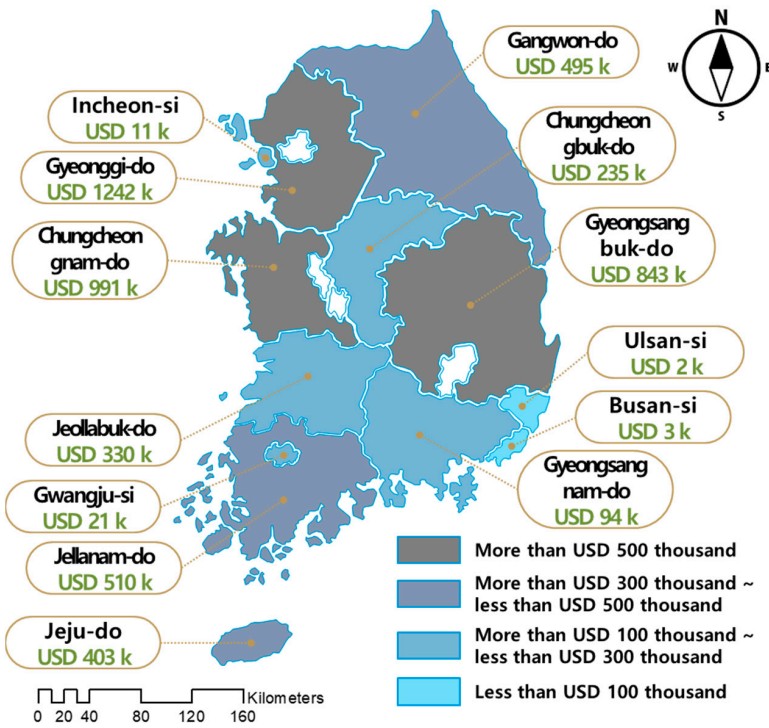

**Figure 3.** Recovery cost from agricultural drought by province in 2018.

In 2018, the regions with agricultural drought recovery cost of USD 500 thousand or more were Gyeonggi Province with USD 1242 thousand, Chungcheongnam-do with USD 991 thousand, and Gyeongsangbuk-do with USD 843 thousand. The recovery cost for each parameter was USD 439 thousand for agricultural drought support, USD 2303 thousand for water support, and USD 2438 thousand for equipment and labor. The 2018 agricultural drought recovery cost by province are shown in Table 12.

**Table 12.** Recovery cost from agricultural drought by province and factor in 2018.

| Si-do | Recovery Cost (Thousand Dollar) | | | |
|---|---|---|---|---|
| | Agriculture Drought Support | Water Support | Equipment and Manpower Support | Sum |
| Seoul-si | 0 | 0 | 0 | 0 |
| Busan-si | 0 | 3 | 0 | 3 |
| Daegu-si | 0 | 0 | 0 | 0 |
| Incheon-si | 5 | 3 | 2 | 11 |
| Gwangju-si | 1 | 12 | 8 | 21 |
| Daejeon-si | 0 | 0 | 0 | 0 |
| Ulsan-si | 0 | 0 | 2 | 2 |
| Sejong-si | 0 | 0 | 0 | 0 |
| Gyeonggi-do | 10 | 1119 | 112 | 1242 |
| Gangwon-do | 73 | 105 | 318 | 495 |
| Chungcheongbuk-do | 7 | 87 | 142 | 235 |
| Chungcheongnam-do | 25 | 517 | 448 | 991 |
| Jeollabuk-do | 20 | 82 | 228 | 330 |
| Jellanam-do | 58 | 152 | 300 | 510 |
| Gyeongsangbuk-do | 35 | 65 | 743 | 843 |
| Gyeongsangnam-do | 0 | 25 | 69 | 94 |
| Jeju-do | 205 | 132 | 65 | 403 |
| Average | 439 | 2303 | 2438 | 5180 |

The recovery cost of agricultural drought assistance totaled USD 439 thousand, with the exception of Jeju Island's USD 205 thousand, most of which was less than USD 100 thousand. The provinces that received more than USD 50 thousand in agricultural drought assistance were Gangwon-do with USD 73 thousand and Jeollanam-do with USD 58 thousand.

Water support is composed of dam water, water supply, supplementary water, groundwater, water trucks, and water bags, with a total recovery cost of USD 2303 thousand. The regions with recovery cost of more than USD 100 thousand were Chungcheongnam-do USD 517 thousand, Jeollanam-do USD 152 thousand, Jeju-do USD 132 thousand, and Gangwon-do USD 105 thousand.

Equipment and personnel recoveries totaled USD 2438 thousand. Recovery cost of USD 300 thousand or more were estimated at USD 743 thousand in Gyeongsangbuk-do, USD 448 thousand in Chungcheongnam-do, USD 318 thousand in Gangwon-do, and USD 300 thousand in Jeollanam-do.

### 3.2.2. Calculating Agricultural Drought Recovery Cost for 2019

In this study, we estimate the amount of agricultural drought recovery cost for 17 provinces nationwide in 2019. The recovery data of agricultural drought was utilized from the National Drought Information and Statistics 2019 (2021, joint ministries) [47]. The criteria for calculating the recovery cost of agricultural drought were the same as in 2018. Also, in 2019, as in 2018, only crop damage was counted as drought damage, and the cost of crop replacement was applied to agricultural drought support.

The basis for calculating recovery cost for agricultural drought assistance, water assistance, and equipment and labor assistance for the 2019 agricultural drought by state is shown in Table 13. Agricultural drought assistance is based on the 2019 standard for large

leek crops, based on the damage calculation. Water assistance and equipment and labor assistance are based on the same criteria as in 2018.

**Table 13.** Calculation standard for agricultural drought recovery cost by province in 2019.

| Agriculture Drought Support | | Water Support | | Equipment and Manpower Support | |
|---|---|---|---|---|---|
| Type | Cost (USD/10 ha) | Type | Cost (USD/100 m³ or Count) | Type | Cost (USD/Count or Person) |
| Crop substitute crops | | Dam water | 4 | Water pump | 231 |
| Paddy | 16 | Tap water | 33 | Water hose | 23 |
| Fields | 362 | Barrage water | 33 | Public official | 64 |
| | | Groundwater | 55 | Military and police | 57 |
| | | Water wagon | 461 | General public | 46 |
| | | Water bag | 55 | | |

In 2019, agricultural drought recovery occurred in 6 out of 17 provinces: Incheon Metropolitan City, Chungcheongnam-do, Chungcheongnam-do, Jeollabuk-do, Gangwon-do, Chungcheongbuk-do, and Gyeongsangbuk-do. Compared to 2018, the number of recovery areas decreased by about 50%. The recovery cost of agricultural drought recovery in 2019 was USD 218,000, with the most affected province being Gyeongsangbuk-do (USD 76,000) and the least affected province being Gangwon-do (USD 2000). Figure 4 shows the agricultural drought recovery status by province.

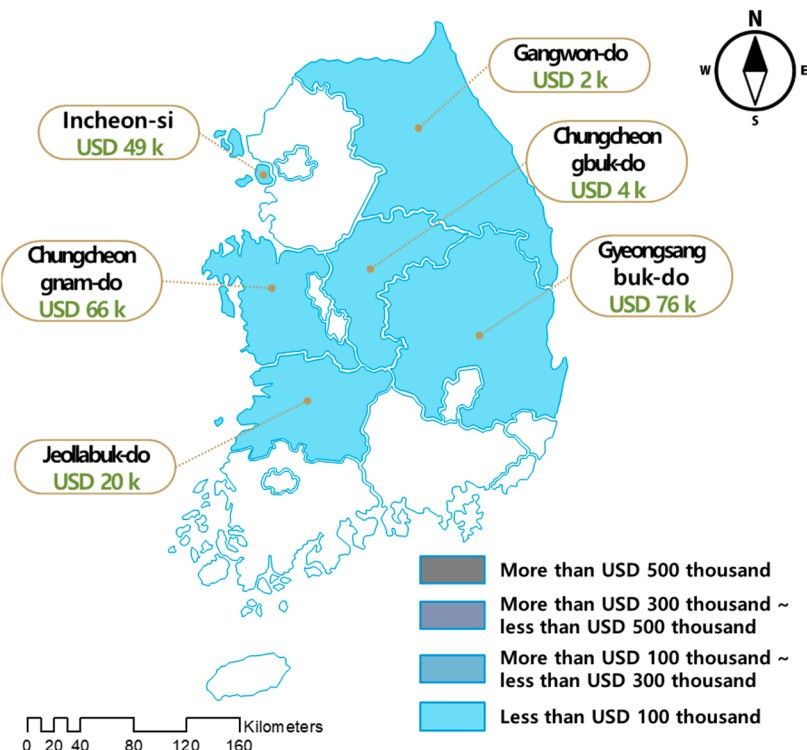

**Figure 4.** Recovery cost from agricultural drought by province in 2019.

The 2019 agricultural drought was assessed as a smaller recovery cost than the 2018 drought not only in terms of damage cost but also in terms of recovery cost. The recovery cost was based on agricultural drought assistance and water assistance only, and equipment and personnel assistance were not utilized. These indicators suggest that the damage of the 2019 agricultural drought was minimal and that the disaster was mitigated by the response to water distribution alone.

In 2019, the regions with agricultural drought recovery costs of more than USD 40,000 were calculated as Gyeongsangbuk-do USD 76,000, Chungcheongnam-do USD 66,000, and Incheon Metropolitan City USD 49,000. The recovery cost for each parameter is USD 12,000 for agricultural drought support and USD 206,000 for water support. The agricultural drought recovery amount by province in 2019 is shown in Table 14.

**Table 14.** Recovery cost from agricultural drought by province and factor in 2019.

| Si/Do | Recovery Cost (Thousands of USD) | | | |
|---|---|---|---|---|
| | Agriculture Drought Support | Water Support | Equipment and Manpower Support | Sum |
| Seoul-si | 0 | 0 | 0 | 0 |
| Busan-si | 0 | 0 | 0 | 0 |
| Daegu-si | 0 | 0 | 0 | 0 |
| Incheon-si | 1 | 48 | 0 | 49 |
| Gwangju-si | 0 | 0 | 0 | 0 |
| Daejeon-si | 0 | 0 | 0 | 0 |
| Ulsan-si | 0 | 0 | 0 | 0 |
| Sejong-si | 0 | 0 | 0 | 0 |
| Gyeonggi-do | 0 | 0 | 0 | 0 |
| Gangwon-do | 2 | 0 | 0 | 2 |
| Chungcheongbuk-do | 0 | 4 | 0 | 4 |
| Chungcheongnam-do | 8 | 58 | 0 | 66 |
| Jeollabuk-do | 0 | 20 | 0 | 20 |
| Jellanam-do | 0 | 0 | 0 | 0 |
| Gyeongsangbuk-do | 0 | 76 | 0 | 76 |
| Gyeongsangnam-do | 0 | 0 | 0 | 0 |
| Jeju-do | 0 | 0 | 0 | 0 |
| Average | 12 | 206 | 0 | 218 |

Agricultural drought assistance totaled USD 12,000, with USD 8000 for Chungcheongnam-do, USD 2000 for Gangwon-do, and USD 1000 for Incheon Metropolitan City. Water support is composed of dam water, water supply, supplementary water, groundwater, water delivery vehicles and water bags, with a total recovery cost of USD 206,000. In the regions where the recovery cost was over USD 40,000, Gyeongsangbuk-do was estimated at USD 76,000, Chungcheongnam-do at USD 58,000, and Incheon Metropolitan City at USD 48,000.

## 4. Discussion

This study analyzed the economic impact of agricultural drought in terms of damage cost and recovery cost. The direct damage caused by agricultural drought and the recovery cost to mitigate the damage cost were quantitatively analyzed. The results of the study suggest the importance of effective disaster management and policy development by developing an analytical methodology for the damage cost and recovery cost of agricultural drought and providing economic losses based on historical damage data. However, most previous studies were conducted on risk, prediction models, policies, and water economics rather than calculating quantitative damage from drought. Only a few studies have applied economic impacts to predict losses in the GDP concept.

Studies on drought damage have analyzed the economic impact of drought in various countries. Economic impacts of climate-change-induced drought have been analyzed in Australia, the United States, and Spain, with projected losses ranging from 0.04% to 9% of GDP [13]. Malawi, Africa assessed the economic impacts using a general equilibrium model and found that 1.7% of GDP was lost annually [15]. In addition, In Iran, the economic impact of agricultural drought was estimated to be USD 1.605 billion, or 4.4% of GDP and inflation in other industries [27]. Some studies have estimated the cost of restoration of indirect damages from agricultural water contamination that are not directly attributable to drought [38]. It was estimated that if water use were reduced by 42% due to an increase

in pollutants in the Ebro River in Spain, damage of USD 53 million to USD 171 million would occur.

Some studies do not estimate quantitative drought damage but present qualitative effects in terms of percentage or probability of drought occurrence [28,32,33,39]. Most of the drought damage estimates using drought indexes and hydrological models suggest reducing qualitative impacts or improving drought management. Drought indices provide probability curves and frequency of crop losses due to drought [28,39]. Some studies have considered two drought indices to predict the predicted yield of crops under agricultural drought or provide quantitative revenue losses as a percentage [32,33]. However, damage estimation using drought indices and hydrological models has limitations in application to quantitative policy applications such as regional water distribution, manpower input, and food distribution for disaster management.

Existing studies on calculating the cost of damage from agricultural drought presented the economic cost as a percentage of GDP or estimated the scale of damage from pollutants. However, the agricultural drought damage cost and recovery cost estimates in this study provide quantitative estimates of the actual damage. It is expected to be more useful for policy proposals and budget allocations for disaster management where actual budgets are involved. In addition, it is possible to calculate a comprehensive cost of agricultural drought damage that considers not only economic losses but also recovery costs. In addition, we calculated the amount of money to recover damages for agricultural drought. The recovery equation only considers recovery factors for damages and excludes indirect impacts.

Although a technique for calculating the damage costs and recovery costs from agricultural drought has been developed, it also has limitations, as the damage factors have not been investigated first. For these missing damage factors, additional data must be secured through cooperation with relevant ministries. In addition, such missing data are expected to cause errors in calculating the amount of damage and recovery that each ministry wishes to confirm. It has limitations in that it only covers direct damage from agricultural drought and does not consider indirect effects. For future research, we would like to propose a methodology that considers both direct and indirect damage by adding indirect impacts to the current methodology for analyzing agricultural drought damage cost and recovery cost.

In this study, we developed an analysis methodology for damage cost and recovery cost based on the first official agricultural drought damage history in Korea and estimated the quantitative regional damage by calculating the damage and recovery for 2018 and 2019. However, it was difficult to apply accurate estimation criteria for the depth and accuracy of agricultural drought due to the difficulty of utilizing data provided by various ministries or organizations. In addition, there are regional limitations in providing damage data on agricultural drought and basic data for calculating damage costs and recovery costs. In Korea, administrative districts are divided into 227 regions at the sigungu level, but most drought damage data are provided at the level of the 17 si/do. Of course, the scale of damage from drought occurs at the si/do level rather than at the sigungu level. However, drought forecasts, warnings and responses are analyzed at the sigungu level, which limits regional uniformity. In the future, research is needed on regional integration of analysis of drought prevention and preparation and damage estimates for response and recovery.

## 5. Conclusions

Quantitative analysis of drought is an area where various studies are being conducted in many countries. In particular, efforts are being made to estimate the exact amount of damage due to agricultural drought, which is the greatest effect of drought. However, in most countries, accurate estimates of drought damage and recovery costs have limitations when applied in practice. Therefore, most insurance companies' insurance rates are applied to estimate the amount of damage. However, in securing a budget for national policy, a clear estimate of the scale of damage for disaster prevention, preparation, response, and recovery is necessary. Therefore, in this study, we developed a methodology to calculate the damage and recovery cost of agricultural drought applicable to Korea.

In this study, we developed a methodology for analyzing agricultural drought damage cost and recovery cost in South Korea. The economic impacts of the 2018 and 2019 agricultural droughts were analyzed for 17 provinces across the country. The damages of agricultural drought were calculated for rice fields and fields that suffered direct losses in 2018 and 2019. In 2018, the total value of agricultural drought damages was USD 4.438 million, with the most affected area being Jeju Island and the least affected area being Gwangju City. In 2019, the total damage costs were USD 286,000, a significant decrease from the previous year, with the most affected area being Gangwon Province. The damage cost varied depending on the severity of the drought, agricultural dependence, and crops grown in each region.

In 2018, the total amount of agricultural drought recovery cost was USD 5.180 million, with the largest affected area being Gyeonggi Province and the smallest being Ulsan City. In 2019, the total recovery cost was USD 218,000, a significant decrease from the previous year, and the largest affected area was Gyeongsangbuk-do. The analysis of the recovery cost shows that various resources and support for resuming agricultural activities after an agricultural drought are costlier than the damage.

The results of this study show that agricultural drought has a significant impact on the region. Therefore, it is urgent to develop strategies to prepare for agricultural droughts, including improved drought forecasting and monitoring systems, efficient water management and water distribution systems, and disaster assistance policies.

This study aims to analyze the economic losses caused by agricultural drought and provide important basic data for efficient recovery and response systems. In addition, the quantitative analysis of agricultural drought losses emphasizes their usefulness in the formulation of drought-related policies and response strategies and raises the need for continued research and data collection.

**Author Contributions:** Conceptualization, Y.S. and J.J.; methodology, Y.S.; software, J.J.; validation, Y.S., H.K. and M.P.; formal analysis, M.P.; investigation, Y.S.; resources, Y.S.; data curation, Y.S.; writing—original draft preparation, Y.S.; writing—review and editing, J.J.; visualization, H.K.; supervision, M.P.; project administration, M.P.; funding acquisition, Y.S. All authors have read and agreed to the published version of the manuscript.

**Funding:** This paper was researched under the Intramural Research Support Project of Hanseo University in 2023.

**Data Availability Statement:** Data is contained within the article.

**Conflicts of Interest:** The authors declare no conflicts of interest.

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
