# Peer review of "Development and Applicability Evaluation of Damage Scale Analysis Techniques for Agricultural Drought"

_water, doi:10.3390/w16101342_

Round 1
Reviewer 1 Report
Comments and Suggestions for Authors
Overall this is a very important topic to discuss. Flooding and risks associated with flooding are becoming a bigger reality as we deal with climate concerns. What is not clear to me is the literature about communicating risk and how that might inform better understanding of the risks the researchers measured. This information is valuable but what is needed here is a deeper understanding of the "Why" this information is valuable to readers, practitioners and scientists. This could be included in the introduction but should also be revisited at the end of the paper.
Another concern I have is the application of the maps in that respect. I think the authors gathered a lot of great data, but I would like to see a summary that explains why the maps are useful for looking at this data, and how this study builds on the practice of risk assessment, specifically in reference to the maps and what they communicate.
Comments on the Quality of English LanguageOverall this is a very well written paper. Adding a few sentences to summarize the work in simpler terms at the end of the paper, might help the paper reach a broader audience and engage a different group of scholars outside economics.
Author Response
Thank you for the reviewer's comments. I have written responses to reviewer comments and revised paper pages. The reviewer's corrections are marked in blue. Again, thanks for the comments, we made the corrections with the utmost care.
comments
- Overall this is a very important topic to discuss. Flooding and risks associated with flooding are becoming a bigger reality as we deal with climate concerns. What is not clear to me is the literature about communicating risk and how that might inform better understanding of the risks the researchers measured. This information is valuable but what is needed here is a deeper understanding of the "Why" this information is valuable to readers, practitioners and scientists. This could be included in the introduction but should also be revisited at the end of the paper.
â–º Modified according to reviewer's opinion. (on page 16)
- Another concern I have is the application of the maps in that respect. I think the authors gathered a lot of great data, but I would like to see a summary that explains why the maps are useful for looking at this data, and how this study builds on the practice of risk assessment, specifically in reference to the maps and what they communicate.
â–º Modified according to reviewer's opinion. (on page 16)
- Overall this is a very well written paper. Adding a few sentences to summarize the work in simpler terms at the end of the paper, might help the paper reach a broader audience and engage a different group of scholars outside economics.
â–º It's the same answer as number 1. Modified according to reviewer's opinion. (on page 16)
Reviewer 2 Report
Comments and Suggestions for Authors
This study developed an analytical methodology to quantitatively assess the economic damage of agricultural drought and estimated the damage of agricultural drought in 2018 and 2019 for the Republic of Korea While the discussion lacks in-depth analysis and interpretation of the findings. Overall, I recommend this MS to be published after a major revision.
1. The introduction is somewhat confusing in structure and lacks clear segmentation and organization.
2. The structure of the discussion section is loose, and the content is awkward to switch between different topics. Discussions can be made more logical and coherent by better organizing paragraphs and information.
3. The discussion lacks in-depth analysis and interpretation of the findings. It is suggested that the results should be further explored and the relationship between the results and the existing literature should be explored.
4. The number of tables is too large, it is recommended to change some of the result tables to graphs for clearer representation
Comments on the Quality of English Languagenone
Author Response
Thank you for the reviewer's comments. I have written responses to reviewer comments and revised paper pages. The reviewer's corrections are marked in red. Duplicate opinions are marked in blue and purple. Again, thanks for the comments, we made the corrections with the utmost care.
comments
- 1. The introduction is somewhat confusing in structure and lacks clear segmentation and organization.
â–º Modified according to reviewer's opinion. (on page 1-2)
- The structure of the discussion section is loose, and the content is awkward to switch between different topics. Discussions can be made more logical and coherent by better organizing paragraphs and information.
â–º Modified according to reviewer's opinion. (on page 15-16)
- The discussion lacks in-depth analysis and interpretation of the findings. It is suggested that the results should be further explored and the relationship between the results and the existing literature should be explored.
â–º Modified according to reviewer's opinion. (on page 15-16)
- The number of tables is too large, it is recommended to change some of the result tables to graphs for clearer representation.
â–º Based on the reviewer's opinion, we have modified the figures for Tables 7, 8, 9, 10, 12, and 14. However, the data had the same values or the differences in data values were large, making clear confirmation difficult. So, we presented it in the same table as the present.
Reviewer 3 Report
Comments and Suggestions for Authors
In this article, the author proposes a statistical method to estimate the damage cost and recovery cost of agricultural drought in South Korea. The method is evaluated based on historical agricultural drought disasters that occurred in 2018 and 2019. However, the overall structure and organization of the article have some issues. The author can improve the quality of the manuscript by addressing the following problems:
1. The author should reorganize the introduction to clarify the research background, ideas, and objectives. Merge the first and second paragraphs for a comprehensive introduction to the study's context and significance. Combine paragraphs four through seven into a new paragraph to systematically present relevant research findings and trends. This restructuring will make the article more concise, logical, and coherent, enhancing the reading experience and helping readers grasp the research focus and direction.
2. The text lacks reference support in discussing the impacts of drought on food security, productivity, and economic losses, as well as the expected increase in agricultural damage due to climate change (Line 31-38). Similar issues are also present in the first, second, and third paragraphs of the introduction, as well as in the discussion section.
3. The references mentioned in paragraphs 5, 6, and 7 of the introduction are not appropriately categorized, which could make it difficult for readers to understand the core content of these references. For example, in lines 58-63 on page 2, "Hydro-economic modeling has been used to assess the impacts of agricultural droughts in Spain, the United States, and China. They also proposed water management and response policies to minimize economic losses [25-26,28,29-31]. In Nigeria, Spain, and Kenya, the economic value of drought forecasting information was assessed and policy and technical measures were proposed to enhance farmers' adaptive capacity [33-34]." The author is advised to group and explain the references based on regions or article themes to help readers better understand and track the specific content of each case.
4. The sentence in lines 64-66 on page 2 contains a grammatical error: "The development of drought prediction models utilized precipitation data and climate models for the UK and South Africa. Drought frequency and intensity were predicted and the impact on agricultural productivity was assessed [35-37]." The correct restructuring is: "The development of drought prediction models utilized precipitation data and climate models for the UK and South Africa. These models predicted both the frequency and intensity of droughts, which were then evaluated for their impact on agricultural productivity [35-37]." This modification makes the sentence structure clearer and the logic more coherent.
5. Table 3 and Table 5 in the Materials and Methods section of Chapter Two have highly overlapping content. It is suggested to merge them into a single unified table to improve the conciseness of the document.
6. The paper only used data from historical record manuals. However, the description regarding 2018 and 2019 (as mentioned in lines 211-212 on page 7) may be misleading. The author should provide more details about the data sources and explain why this estimation method was used, as well as how the data's validity was ensured.
7. The discussion section lacks depth, and the author needs to strengthen the in-depth analysis of the research results, ensuring that each viewpoint is adequately supported. Additionally, reflecting on and prospecting existing achievements, comparing and analyzing them with the work of other scholars, can highlight the importance and innovation of this study.
8. The original language expression needs further optimization, especially the words used to connect logical relationships. For example, in lines 13-14 on page 1, "Agricultural drought is not only a natural disaster, but also leads to food security threats and reduced economic activities such as decreased productivity." The author may want to emphasize the multiple negative impacts of drought, but the sentence's fluency and expression are not effective. It is recommended to revise it.
9. It is recommended that the author include an overview map of the study area in the paper, accompanied by corresponding text descriptions detailing the region's topography, geomorphological features, and climate conditions. This will help readers better understand the research context and provide necessary geographic information for subsequent analysis.
10. This article falls within the realm of conventional research, primarily relying on basic data analysis methods, lacking novelty and unique insights. As a result, its academic value is relatively limited.
Author Response
Thank you for the reviewer's comments. I have written responses to reviewer comments and revised paper pages. The reviewer's corrections are marked in purple. Duplicate opinions are marked in blue and red. Again, thanks for the comments, we made the corrections with the utmost care.
comments
- 1. The author should reorganize the introduction to clarify the research background, ideas, and objectives. Merge the first and second paragraphs for a comprehensive introduction to the study's context and significance. Combine paragraphs four through seven into a new paragraph to systematically present relevant research findings and trends. This restructuring will make the article more concise, logical, and coherent, enhancing the reading experience and helping readers grasp the research focus and direction.
â–º Modified according to reviewer's opinion. (on page 1-2)
- The text lacks reference support in discussing the impacts of drought on food security, productivity, and economic losses, as well as the expected increase in agricultural damage due to climate change (Line 31-38). Similar issues are also present in the first, second, and third paragraphs of the introduction, as well as in the discussion section.
â–º Modified according to reviewer's opinion. (on page 1-2)
- The references mentioned in paragraphs 5, 6, and 7 of the introduction are not appropriately categorized, which could make it difficult for readers to understand the core content of these references. For example, in lines 58-63 on page 2, "Hydro-economic modeling has been used to assess the impacts of agricultural droughts in Spain, the United States, and China. They also proposed water management and response policies to minimize economic losses [25-26,28,29-31]. In Nigeria, Spain, and Kenya, the economic value of drought forecasting information was assessed and policy and technical measures were proposed to enhance farmers' adaptive capacity [33-34]." The author is advised to group and explain the references based on regions or article themes to help readers better understand and track the specific content of each case.
â–º Modified according to reviewer's opinion. (on page 1-2)
- The sentence in lines 64-66 on page 2 contains a grammatical error: "The development of drought prediction models utilized precipitation data and climate models for the UK and South Africa. Drought frequency and intensity were predicted and the impact on agricultural productivity was assessed [35-37]." The correct restructuring is: "The development of drought prediction models utilized precipitation data and climate models for the UK and South Africa. These models predicted both the frequency and intensity of droughts, which were then evaluated for their impact on agricultural productivity [35-37]." This modification makes the sentence structure clearer and the logic more coherent.
â–º Modified according to reviewer's opinion. (on page 2)
- Table 3 and Table 5 in the Materials and Methods section of Chapter Two have highly overlapping content. It is suggested to merge them into a single unified table to improve the conciseness of the document.
â–º I tried to integrate Tables 3 and 5 according to the reviewer's opinion, but I had to read pages 4 to 6 repeatedly to understand the paper. So, we decided to keep the format as it is now. (on page 4,6)
- The paper only used data from historical record manuals. However, the description regarding 2018 and 2019 (as mentioned in lines 211-212 on page 7) may be misleading. The author should provide more details about the data sources and explain why this estimation method was used, as well as how the data's validity was ensured.
â–º Modified according to reviewer's opinion. (on page 7,9)
- The discussion section lacks depth, and the author needs to strengthen the in-depth analysis of the research results, ensuring that each viewpoint is adequately supported. Additionally, reflecting on and prospecting existing achievements, comparing and analyzing them with the work of other scholars, can highlight the importance and innovation of this study.
â–º Modified according to reviewer's opinion. (on page 15-16)
- The original language expression needs further optimization, especially the words used to connect logical relationships. For example, in lines 13-14 on page 1, "Agricultural drought is not only a natural disaster, but also leads to food security threats and reduced economic activities such as decreased productivity." The author may want to emphasize the multiple negative impacts of drought, but the sentence's fluency and expression are not effective. It is recommended to revise it.
â–º Modified according to reviewer's opinion. (on page 1-2)
- It is recommended that the author include an overview map of the study area in the paper, accompanied by corresponding text descriptions detailing the region's topography, geomorphological features, and climate conditions. This will help readers better understand the research context and provide necessary geographic information for subsequent analysis.
â–º In cases where analysis of meteorological or topographical data such as drought index and hydrological modeling is included, I think supplementation according to the reviewer's opinion is necessary. However, this study analyzed the quantitative impact on the scale of damage after it occurred, and geographic information was not applied. Therefore, this study will present the impact of geographic information in future research.
- This article falls within the realm of conventional research, primarily relying on basic data analysis methods, lacking novelty and unique insights. As a result, its academic value is relatively limited.
â–º We apologize for the reviewer's comments. However, in many countries, research on quantitatively estimating drought damage data or damage amount has been an area of continuous research for several years. In addition, drought damage is very sensitive data, and the damage data has not been made public, making it difficult to develop a quantitative calculation formula. In addition, the method of calculating the amount of damage from agricultural drought in terms of damage and recovery amount is also a research area with insufficient prior research. We will reflect the reviewers' opinions and strive to make the research more academically valuable in the future.
Round 2
Reviewer 2 Report
Comments and Suggestions for Authors
I think the authors have basically addressed the comments or questions.
Comments on the Quality of English Languagenone
Author Response
Thank you for the reviewer's comments. All comments made in the first round have been revised. Thanks for checking.
Reviewer 3 Report
Comments and Suggestions for Authors
Despite some revisions made by the author, there are still portions of the article that remain unchanged and the language still appears cumbersome. To enhance readability and coherence, I suggest the author further revises and simplifies the language to ensure that readers can easily comprehend and accept the author's viewpoints and research findings.
1. The introduction of the article lacks sufficient supporting references in the first and second paragraphs.
2. The repetition between lines 86-88 and lines 49-52 on page 2 should be removed to ensure clarity in the structure and logic of the article.
3. Equation 1 calculates the damage cost of agricultural drought as the sum of crop, livestock, and fishery damage costs. However, the article only considers crop damage data and does not account for livestock and fishery damage costs. The author should discuss the potential impact of this missing data on the results in the discussion section.
4. The article estimates damage cost of agricultural drought for the years 2018 and 2019 in South Korea despite various data gaps (lines 212-210 on page 7 and lines 260-267 on page 9). The basis for this choice is not mentioned in the text and should be explained.
Comments on the Quality of English LanguagePlease write in a succinct and concise manner.
Author Response
Thank you for the reviewer's comments. I have written responses to reviewer comments and revised paper pages. The reviewer's corrections are marked in blue. Again, thanks for the comments, we made the corrections with the utmost care.
comments
- 1. The introduction of the article lacks sufficient supporting references in the first and second paragraphs.
â–º Modified according to reviewer's opinion. (on page 1)
â–º The author wrote the introduction of the article based on what he considered to be general information after studying various previous studies. Based on the reviewer's opinion, we additionally present a paper that contains the meaning of the content presented in research other than existing previous research.
- The repetition between lines 86-88 and lines 49-52 on page 2 should be removed to ensure clarity in the structure and logic of the article.
â–º Modified according to reviewer's opinion. (on page 2)
- Equation 1 calculates the damage cost of agricultural drought as the sum of crop, livestock, and fishery damage costs. However, the article only considers crop damage data and does not account for livestock and fishery damage costs. The author should discuss the potential impact of this missing data on the results in the discussion section.
â–º Modified according to reviewer's opinion. (on page 16)
- The article estimates damage cost of agricultural drought for the years 2018 and 2019 in South Korea despite various data gaps (lines 212-210 on page 7 and lines 260-267 on page 9). The basis for this choice is not mentioned in the text and should be explained.
â–º Modified according to reviewer's opinion. (on page 7, 9)